# Uncertainty-Aware Training of Neural Networks for Selective Medical Image Segmentation

**Yukun Ding** [1]                         YDING5@ND.EDU

**Jinglan Liu** [1]                          JLIU16@ND.EDU

**Xiaowei Xu**[2]                          XXU8@ND.EDU

**Meiping Huang**[2]                    HUANGMEIPNG@126.COM

**Jian Zhuang**[2]                   ZHUANGJIAN5413@TOM.COM

**Jinjun Xiong**[3]                      JINJUN@US.IBM.COM

**Yiyu Shi**[1]                           YSHI4@ND.EDU

[1] *University of Notre Dame*

[2] *Guangdong General Hospital*

[3] *IBM Thomas J. Watson Research Center*

## Abstract

State-of-the-art deep learning based methods have achieved remarkable performance on medical image segmentation. Their applications in the clinical setting are, however, limited due to the lack of trustworthiness and reliability. Selective image segmentation has been proposed to address this issue by letting a DNN model process instances with high confidence while referring difficult ones with high uncertainty to experienced radiologists. As such, the model performance is only affected by the predictions on the high confidence subset rather than the whole dataset. Existing selective segmentation methods, however, ignore this unique property of selective segmentation and train their DNN models by optimizing accuracy on the entire dataset. Motivated by such a discrepancy, we present a novel method in this paper that considers such uncertainty in the training process to maximize the accuracy on the confident subset rather than the accuracy on the whole dataset. Experimental results using the whole heart and great vessel segmentation and gland segmentation show that such a training scheme can significantly improve the performance of selective segmentation.

**Keywords:** Image Segmentation, Selective Segmentation, Uncertainty Estimation.

## 1. Introduction

Deep neural networks (DNNs) have greatly reduced human efforts needed to segment medical images. However, their adoption in clinical procedures is relatively slow. One of the foremost reasons is the lack of trustworthiness and reliability, which is critical for medical applications. Given DNN models' fundamental dependency on sufficient annotated samples and poor generalization on unseen data (Wang et al., 2018a), it is hard to expect that they alone can provide trusted and reliable segmentation in the near future (Joskowicz et al., 2019). A much more likely path forward is to rely on human-machine cooperation such that DNN models process easy instances in a confident and reliable manner and refer difficult ones with uncertainty to experienced radiologists, which is also known as *selective segmentation* (Nair et al., 2018; Sander et al., 2019).

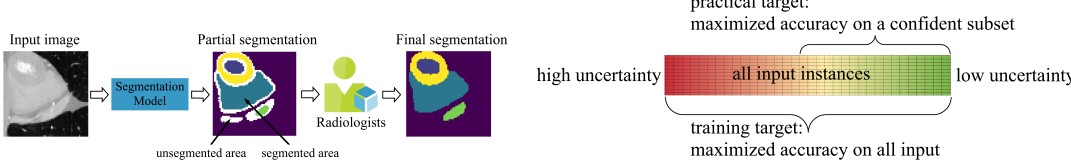

($a$) Process of selective segmentation     ($b$) The training target and practical target

Figure 1: Concept illustration of selective segmentation.

The process of selective segmentation is shown in Figure 1($a$). The DNN model will process the input image first. Instead of letting the model segment the whole image regardless of the difficulty of instances, the model will selectively segment part of the input image, on which it is confident to make decisions (i.e. a confident subset of all inputs). The remaining part will be highlighted as unsegmented which is the white area in Figure 1($a$) and referred to radiologists to complete the final segmentation. The coverage of the model is defined as the percentage of the segmented area by the model accordingly.

The accuracy of DNN models in the context of selective segmentation should be evaluated differently compared with the conventional segmentation. For conventional segmentation, a model makes predictions on all data and the *training target* is maximizing the accuracy on all instances in the training set. In contrast, for selective segmentation, what will be used in the inference stage is only the confident subset of instances, where the model makes confident predictions. This means that the *practical target* is no longer maximizing the segmentation accuracy over the entire inputs but rather on a subset, which is to be determined. An illustration is shown in Figure 1($b$). Therefore, it is sub-optimal to still train the model with the same objective functions as in normal segmentation tasks. However, while all existing neural network-based methods differ in the way they extract and process the information from the network, they are trained with conventional objective functions designed for the conventional accuracy measure on the entire training set (Nair et al., 2018; Sander et al., 2019), leading to a gap between the *training target* and the *practical target*. A detailed review of related work is provided in Appendix A.

We narrow this gap by developing a new method for selective segmentation that considers the joint effect of uncertainty estimation and prediction to directly optimize for the *practical target* in the training process. More importantly, we show that the *practical target* can be connected to the *training target* by a novel uncertainty target. This enables us to transform the optimization for the *training target* to optimization for the *practical target* by adding an uncertainty loss term that accounts for the uncertainty target in the final loss function. This is powerful as it does not require any special network architecture and is universally applicable to all existing DNN models with little implementation effort. As such, our method can be easily adopted in a plug-and-play manner to most segmentation frameworks and enjoy all the benefits of new techniques for general segmentation tasks. Experimental results show that, our method significantly improves the selective segmentation performance. To the best of our knowledge, this is the first work that considers uncertainty in the training process for selective segmentation.

## 2. Method

In this section, we first formally define the selective image segmentation problem. Then we introduce our key observation and present a novel uncertainty estimation target that connect the *practical target* and the *training target*. Finally we discuss the application of the proposed objective in uncertainty-aware training and the computation overhead.

### 2.1. Problem Setting

Without any assumption on how the quantified uncertainty score is obtained (where various methods exist and can be applied), for each instance $x_i \in X$ (pixel or voxel in the case of 2D and 3D medical images, respectively), a model $f$ produces the predicted class $\hat{y}_i \in \hat{Y}$ and a quantified uncertainty scalar $u_i \in U$ as:

$$\hat{Y}, U = f(X) \tag{1}$$

The correctness score $s_i = 1$ if $y_i = \hat{y}_i$, and $s_i = 0$ if $y_i \neq \hat{y}_i$, where $y_i \in Y$ is the ground truth label of $x_i$. Note that $\hat{Y}$ is not necessarily the same as $Y$. We further define $U_w = \{u_i | s_i = 0\}$ and $U_c = \{u_i | s_i = 1\}$ as the sets of uncertainties of wrong predictions and correct predictions, respectively.

When a threshold $t$ is applied to the uncertainty measure $u_i$, it splits instances into two sets as $X_l = \{x_i | u_i \leq t\}$ (with the corresponding $\hat{Y}_l = \{\hat{y}_i | u_i \leq t\}$) and $X_h = X - X_l$ (with the corresponding $\hat{Y}_h = \hat{Y} - \hat{Y}_l$), where we use minus to represent the set subtraction operation. For selective segmentation inference, instances in $X_l$ will be segmented by the DNN model with label $\hat{Y}_l$, while instances in $X_h$ will be referred to radiologists for manual segmentation.

The coverage of a DNN model with threshold $t$ is defined as $c = \frac{|X_l|}{|X|}$. We denote $\psi_c$ as the instance-wise accuracy at a coverage $c$: $\psi_c = \frac{\sum_{x_i \in X_l} s_i}{|X_l|}$. When $c = 1$, $X_l = X$ and we have $\psi_1 = \frac{\sum_{x_i \in X} s_i}{|X|}$. For any given DNN model, $\psi_c$ and $c$ define an accuracy-coverage curve by varying $c$.

In practice, $\psi_c$ has an increasing trend with decreasing $c$. The specific practical operating point $\{c, \psi_c\}$ on the accuracy-coverage curve can be chosen by radiologists as needed for a desired trade-off between accuracy and coverage. For example, for critical tasks, $c$ can be selected such that $\psi_c$ is higher than a pre-defined accuracy threshold. Statistical methods can be used to guarantee the desired accuracy with high probability on unseen data (Geifman and El-Yaniv, 2017). Clearly, $\psi_1$ is the *training target* which is the aim of training algorithm in all existing work (Nair et al., 2018; Sander et al., 2019), while $\psi_c$ is the pactical target that actually determines the performance of the model in a selective segmentation system.

### 2.2. The Uncertainty Target

While we are clear about the practical target $\psi_c$, it is unclear how to directly optimize $\psi_c$. The actual performance of selective segmentation $\psi_c$, is a combined result of the normal segmentation result and uncertainty estimation. For any fixed segmentation results, the performance of selective segmentation (i.e. $\psi_c$) is maximized if we have a perfect uncertainty

estimation. However, while the optimization for segmentation performance is well-studied, that for uncertainty estimation is not clear.

The optimization for uncertainty estimation rely on an objective function which can be analyzed in the framework of scoring rule (Gneiting and Raftery, 2007; Jolliffe and Stephenson, 2012), which provides an attractive property for optimization problems. A scoring rule is defined as a quantified summary measure for the quality of probabilistic predictions. Denote the true distribution as $q$ and the distribution predicted by a model parameterized by $\theta$ is $p_\theta$. A scoring rule $h$ is a proper scoring rule if $h(p_\theta, q) \leq h(q, q)$ for all $p_\theta$ and $q$. Additionally, $h$ is a strictly proper scoring rule if $h(p_\theta, q) \leq h(q, q)$ with equality if and only if $p_\theta = q$. A proper scoring rule can be used as a loss function to find $\arg\max_\theta h(p_\theta, q)$.

The uncertainty estimation can be considered as such a probabilistic prediction problem. For general uncertainty estimation of neural networks (DeVries and Taylor, 2018a; Hendrycks and Gimpel, 2016; Lakshminarayanan et al., 2017), the normalized uncertainty estimation $u_i \in [0, 1]$ is expected to be the probability of the prediction being wrong and $1 - u_i$ is the probability of the prediction being correct. Then there is a ground truth distribution $q$ to be approximated. It is shown that some commonly used loss functions such as the softmax cross-entropy and Brier score are strictly proper scoring rule (Lakshminarayanan et al., 2017). In other words, a model trained with softmax cross-entropy or Brier score are encouraged to recover the true uncertainty distribution $q$. This is why the maximum softmax (Hendrycks and Gimpel, 2016) can be a good uncertainty estimation for DNNs.

In selective segmentation, we determine whether a instance is high-uncertainty or low-uncertainty based on a variable threshold $t$. For given prediction $\hat{Y}$ and coverage $c$, the model performance $\psi_c$ is only affected by the relative ranking of correction prediction and wrong predictions. As such, what we only need is that wrong predictions are assigned with higher uncertainty than correct predictions, rather than having correct predictions be assigned with $u_i = 1$ and having wrong predictions be assigned with $u_i = 0$.

**Observation**: *For the uncertainty estimation in selective segmentation, we do not need a strictly proper scoring rule that tries to recover the actual distribution $q$.*

Motivated by such a key observation, we look into a measure of uncertainty estimation quality specialized for selective segmentation and denoted by $\gamma$. The following theorem summarizes the desired property of $\gamma$, which facilitates the optimization for $\psi_c$ and justifies our choice. The proof is given in Appendix B.

**Theorem 1** *Denote $\gamma = \frac{\sum_{x_i \in X_h}(1 - s_i)}{|X_h|}$, then we have the following properties about $\psi_c$, $\psi_1$, and $\gamma$ for $c \in (0, 1]$:*

*(i) $\gamma$ is a proper scoring rule but not a strictly proper scoring rule for uncertainty estimation.*

*(ii) $\psi_c = \frac{\psi_1 - (1 - \gamma)(1 - c)}{c}$, s.t. $(1 - \gamma)(1 - c) \in [\psi_1 - c, \psi_1]$.*

*(iii) $\frac{\partial \psi_c}{\partial \gamma} > 0$ and $\frac{\partial \psi_c}{\partial \psi_1} > 0$ for any $\gamma$ and $c$.*

Specifically, $\gamma$ denotes the proportion of wrongly predicted instances among those instances referred to radiologists for any coverage $c$. Now we discuss why these properties are important.

For property $(i)$, as a proper scoring rule, $\gamma$ has the advantage that it minimizes potential conflict with the optimization for $\psi_1$. For example, when maximum softmax probability is used for uncertainty estimation, the uncertainty estimation shares the same output with the normal prediction for every instance. A scoring rule that prefers bad distribution prediction could impede the optimization for $\psi_1$ heavily as $\psi_1$ relies on the approximation of $q$ by $p_\theta$. As a non-strictly proper scoring rule, it avoids the redundant objective of recovering the true uncertainty distribution $q$. Specifically, in order to maximize $\psi_c$, we only need to maximize $\gamma$ which is an easier task compared with recovering the true distribution $q$. Directly maximizing $\gamma$ instead of focusing on $q_\theta$ gives the model more flexibility. As a result, we can expect a higher $\gamma$ and then a higher $\psi_c$.

From property $(ii)$, we can see that for any coverage c, $\psi_c$ is fully determined by $\psi_1$ and $\gamma$. For property $(iii)$, because the partial derivatives of $\psi_c$ with respect to $\psi$ and $\gamma$ are always positive, maximizing $\gamma$ or maximizing $\psi_1$ leads to maximized $\psi_c$ given that the other is not affected. By relaxing the assumption that the other is fixed, we can see that the key to the optimal $\psi_c$ is finding a good trade-off between $\psi_1$ and $\gamma$. Therefore, we propose to maximize $\gamma$ and $\psi_1$ simultaneously in order to maximize $\psi_c$. Compared to other possible formulations, the key advantage of this decomposition is that we are making minimal modification to the conventional optimization method. Therefore, our method is generally applicable and able to leverage any existing techniques for optimizing the training target $\psi_1$ which is important in the application of selective segmentation.

## 2.3. Uncertainty-Aware Training

The discussion above has shown that $\gamma$ is indeed an excellent objective for uncertainty estimation in selective segmentation. However, $\gamma$ is discontinuous and does not provide any useful gradients for the parameters in a neural network. It makes it difficult to use $\gamma$ in the practical model training process. Therefore, we introduce an uncertainty loss as a differentiable proxy to $\gamma$. Specifically, we minimize the following uncertainty loss in the training process,

$$\mathcal{L}_{uncertainty} = \sum_{u_j \in U_w, u_k \in U_c} \max(u_k - u_j + m, 0), \qquad (2)$$

where $m$ is a hyper-parameter that denotes the desired margin to separate $U_w$ and $U_c$. $\mathcal{L}_{uncertainty}$ resembles a margin ranking loss that tries to assign all incorrect prediction with higher uncertainty than all correct predictions i.e. ranking all incorrect predictions higher than correct predictions.

Formally, when $m = 0$, $\gamma$ is maximized over $c \in (0, 1]$ if and only if $\mathcal{L}_{uncertainty}$ is minimized. In order to improve the generalization, a small margin $m$ is used to further separate $U_w$ and $U_c$, which makes minimized $\mathcal{L}_{uncertainty}$ a sufficient but not a necessary condition for maximized $\gamma$. Besides, considering the average value of $\gamma$ when $c \in (0, 1]$, a wrong prediction with lower uncertainty has a higher impact than a wrong prediction with higher uncertainty. This effect is consistent with that in $\mathcal{L}_{uncertainty}$.

It is important to note that $\mathcal{L}_{uncertainty}$ is just an simple but practical approximation to $\gamma$. It is possible to find better optimization proxy and we leave this for future work. As for $\psi_1$, it can be maximized by any conventional training objective functions which is out of the scope of this paper. In order to maximize the overall selective segmentation performance

which is $\psi_c$ for undetermined $c$, we combine the two objectives by a weight $\lambda$ to find a good trade-off between $\gamma$ and $\psi$. The final objective function is given by

$$\mathcal{L}_{u\text{-}seg} = \mathcal{L}_{segmentation} + \lambda\mathcal{L}_{uncertainty}. \tag{3}$$

As we discussed above, the existing segmentation techniques on improving the *training target* is sub-optimal in the selective segmentation scenario becausee of the gap between *training target* and practical target. The proposed method works as a simple yet effective remedy to bridge the gap. We remark that the analysis works for general problems and our method is orthogonal to most state-of-the-art segmentation methods regardless of specific training methods or network architecture. When adopting the uncertainty-aware training scheme to existing segmentation framework, we can directly replace the original loss function $\mathcal{L}_{segmentation}$ with $\mathcal{L}_{u\text{-}seg}$ and run the whole pipeline normally.

Same as the standard stochastic gradient descent, we evaluate and optimize $\mathcal{L}_{uncertainty}$ on each batch. The computation complexity for $\mathcal{L}_{uncertainty}$ is $\mathcal{O}(n^2)$, where $n$ is the number of instances in each batch. For image segmentation, each pixel/voxel is an instance and thus the total number of instances in each batch is fairly big especially on volume data which slows down the training significantly. Therefore, we apply random subsampling to reduce the computation cost. As will be shown in the experimental results, satisfactory accuracy can be achieved with a relatively small sample size.

It is tempting to try a two-stage training scheme to further reduce the overhead. Specifically, one can divide the original training process into two stages. In the first stage, the network is trained with $\mathcal{L}_{segmentation}$ only. After a certain number of epochs we start the second stage where $\mathcal{L}_{u\text{-}seg}$ is used. If the model can adapt from the target $\mathcal{L}_{segmentation}$ to the other target $\mathcal{L}_{u\text{-}seg}$ quickly, we can get similar performance with a lower computation cost. However, in the experiment we find that such assumption is not valid. Detailed results are shown in Appendix D.

## 3. Experiments and Results

### 3.1. Experiments Setup

We first evaluate our proposed method on the Multi-Modality Whole Heart Segmentation (MM-WHS) dataset (Zhuang et al., 2019) with popular backbone network 3D U-Net (Çiçek et al., 2016; Ronneberger et al., 2015). In order to show that the superiority of our method is independent of the power of the network itself, we further validate it on a more challenging dataset from Gland Segmentation Challenge Contest (GlaS) (Sirinukunwattana et al., 2017) using a relatively weak network structure (Ronneberger et al., 2015). We apply uncertainty-aware training with the maximum softmax probability (DeVries and Taylor, 2018b) and the method without our uncertainty loss is the baseline method. More details of the experiments are given in Appendix C. Our code is available at *https://github.com/yding5/Uncertainty-aware-training*.

### 3.2. Results

There are a few hyper-parameters in the methods which can be chosen by a simple grid search. The effect of these hyper-parameters is investigated in Appendix D. The main

results are shown in Table 1. In order to have a quantified metric to evaluate the overall comparisons for the performance of selective segmentation, we use the Area Under Risk-Coverage curve (AURC) following (Ding et al., 2019; Geifman and El-Yaniv, 2017). The Risk-Coverage curve shows the change of Risk with coverage where the Risk is defined as the instance-wise error rate in the image classification scenario. In this paper we compute the risk as $1 - $ Dice to adapt it for segmentation. We also show the distribution of the Dice across all the test images for representative coverage values. From the table we can see that our method consistently improves the AURC and Dice by a significant margin (the mean Dice is increased and the standard deviation is reduced). We also compare the Dice at the 5th percentile of the distribution, which shows that our method can also effectively improve the Dice in most worst cases. We provide detailed image-by-image comparison analysis and the qualitative results below.

Table 1: Quantitative comparison between baseline and our method in terms of AURC (%) mean and standard deviation of Dice across all test images, and Dice at the 5th percentile (5PCTL).

| Dataset | AURC (%) | | Coverage | Dice (%) | | Dice@5PCTL (%) | |
| | Baseline | Ours | | Baseline | Ours | Baseline | Ours |
| --- | --- | --- | --- | --- | --- | --- | --- |
| MM-WHS | 0.936 | **0.810** | 0.95 | 93.86±3.66 | **94.72±2.53** | 83.18 | **89.19** |
| | | | 0.90 | 96.73±2.30 | **97.35±1.54** | 90.55 | **94.64** |
| | | | 0.80 | 98.98±0.79 | **99.21±0.61** | 97.42 | **98.11** |
| | | | 0.70 | 99.64±0.31 | **99.72±0.31** | 98.98 | **99.07** |
| GlaS | 6.981 | **6.031** | 0.95 | 78.62±16.87 | **80.80±16.48** | 35.99 | **39.16** |
| | | | 0.90 | 82.14±14.53 | **84.26±13.87** | 49.77 | **52.46** |
| | | | 0.80 | 87.54±12.31 | **89.37±11.24** | 66.63 | **67.90** |
| | | | 0.70 | 91.45±10.86 | **92.92±9.55** | **75.95** | 74.81 |

**Per-image Comparison.** We plot the per-image comparison between the baseline and our method under different coverage $c$ on MM-WHS in Figure 2. For every input image we compute its coverage $c_1$ and Dice $d_1$ with baseline method and coverage $c_2$ and Dice $d_2$ with our method. The x-axis is the coverage difference $c_2 - c_1$ and the y-axis is the Dice difference $d_2 - d_1$. Note that the $c$ in the caption is the average coverage of all images while each dot represents a specific image with its corresponding coverage and Dice.

The dots in the first quadrant mean our method provides both higher coverage and higher Dice. The dots in the third quadrant mean the opposite. The numbers in the figure indicate the percentages of dots in the corresponding quadrants. The number of dots on the left side of y-axis is approximately the same as that on the right side. This is because the two methods have the same average coverage. On the other hand, the number of dots above the x-axis is significantly more than that below the x-axis. The reason is that our algorithm improves the Dice for most images.

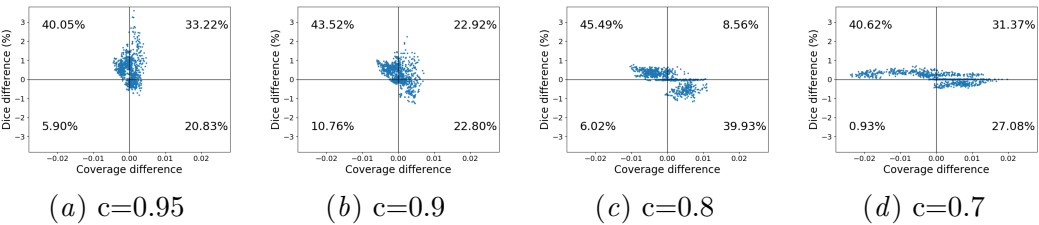

Figure 2: Per-image comparison of Dice and coverage difference under different coverage on MM-WHS.

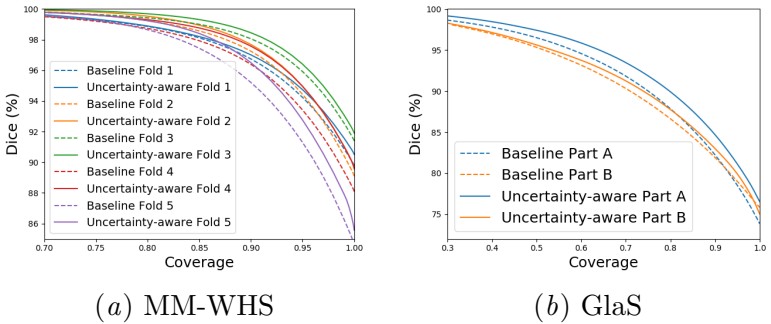

Figure 3: Dice-Coverage curves.

Moreover, the dots in the first quadrant is consistently more than the dots in the third quadrant, which means that our method gets more samples with both higher coverage and accuracy than samples with both lower coverage and lower accuracy. Comparing the results under different coverage, we can see that the Dice difference decreases and the coverage difference increases along with reduced coverage. More results on GlaS are given in Appendix E due to space limitation where we make the same observations. The dots are more sparse for GlaS because of the smaller number of images and higher difficulty of this dataset compared to MM-WHS.

**Dice-Coverage curve.** In order to visualize the overall increased segmentation performance along with more referred instances, we plot Dice-Coverage curves in Figure 3. For MM-WHS, we plot the results of 5-fold cross-validation. For GlaS, we plot the results on the test dataset which has two parts (Sirinukunwattana et al., 2017). In general, our method has equivalent or slightly higher Dice at the full coverage ($c = 1$) compared with the baseline due to the regularization effect of the new loss function which does not penalize heavily for the low confidence wrong prediction. The curves converge when $c$ is too small because it becomes too easy to exclude wrong predictions and thus not shown in the figure. Meanwhile, we observe a clear improvement over a wide range of $c$. We did additional analysis in Appendix F to validate that the model benefits from better uncertainty estimation.

**Qualitative Results.** The qualitative results are shown in Figure 4 where we show representative slices in the region of interest. When the coverage $c$ is 1 that is the normal case for which the baseline is optimized, our method produces similar segmentation results. The

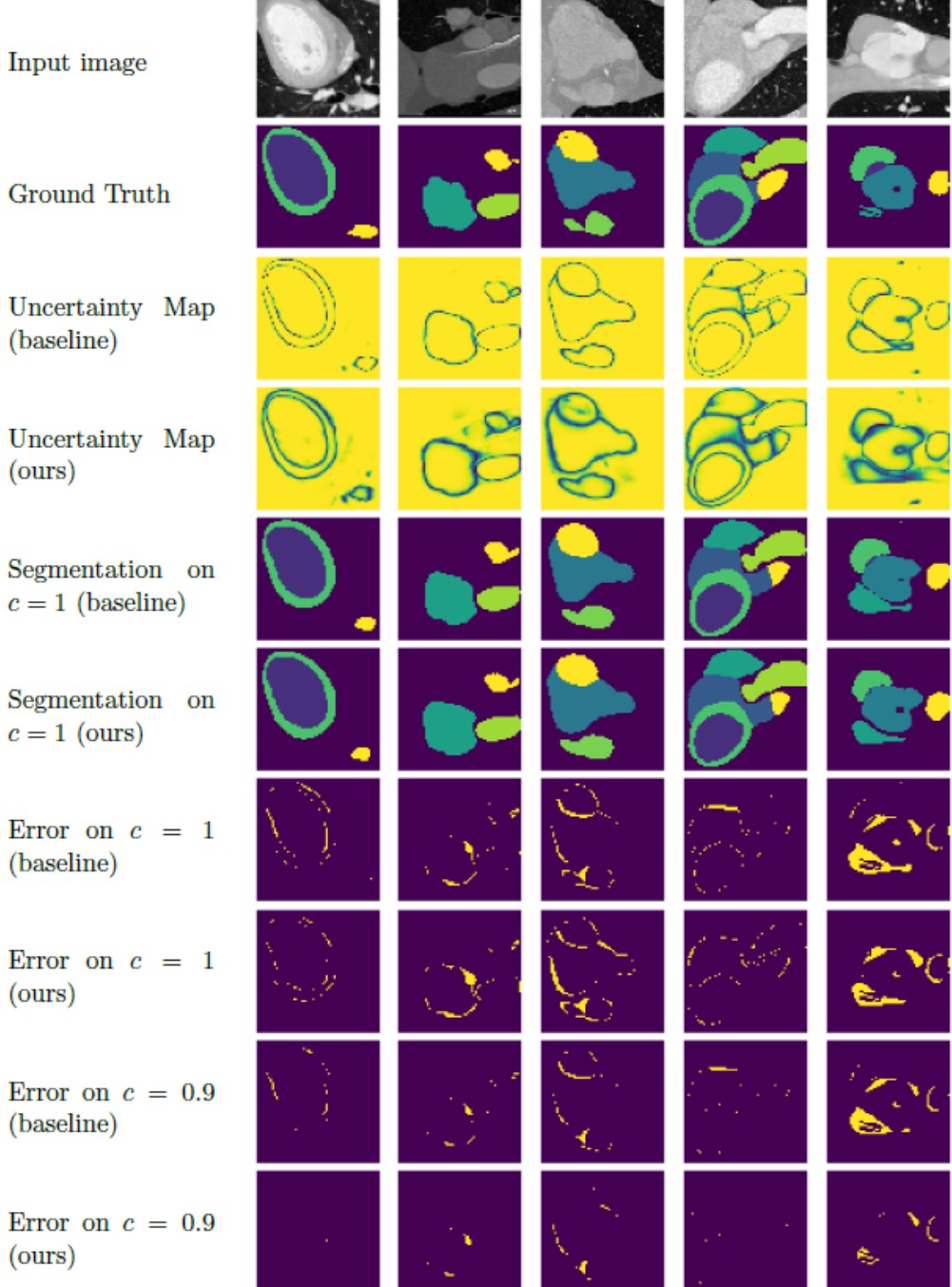

Figure 4: Qualitative comparison with the baseline on MM-WHS. For the error figure, blue means correct segmentation while yellow indicates segmentation error.

original uncertainty map resembles the segmentation boundary because of the ambiguous boundary and the inter-observer variability (Joskowicz et al., 2019; Jungo et al., 2018). Compared to the baseline model, our model's uncertainty map shows more regions with visible uncertainty. The reason is that our training objective penalizes the over-confidence prediction and results in a relatively uniform confidence distribution.

Note that a more uniform distribution does not mean better uncertainty estimation for selective prediction. Only a better ranking of uncertainty leads to better performance when $c < 1$. Besides, it can be seen that the segmentation error at the coverage of 0.9 is greatly reduced compared with $c = 1$ which is the main benefit of selective segmentation.

## 4. Conclusions

In this paper, we develop the first uncertainty-aware training method of neural networks for selective medical image segmentation. Our key observation is that, in a selective segmentation scenario, the *practical target* is different from the conventional *training target* that existing training algorithms are designed for. Motivated by this observation, we proposed an uncertainty-aware training technique that considers uncertainty in the training process to directly maximize the selective segmentation performance in practice, thus closing the gap between the *training target* and the *practical target* in existing methods. The proposed method can be easily adopted into existing frameworks in a plug-and-play manner. Experimental results using whole heart and great vessel segmentation and gland segmentation show that such a training scheme can significantly improve the performance of selective segmentation.

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

## Appendix A. Related Work

### A.1. Uncertainty estimation

There are different ways for the uncertainty estimation of DNNs. A conventional one is approximating Bayesian neural networks by Monte Carlo dropout at inference time (Nair et al., 2018; Shi et al., 2018). Motivated by an interpretation of dropout as ensemble model combination, the use of ensemble for uncertainty estimation is also investigated (Lakshminarayanan et al., 2017; Geifman et al., 2018). In practice, the most popular approach is maximum softmax probability (Hendrycks and Gimpel, 2016), a free by-product of any models that generate probabilities of different classes by a softmax layer. A number of alternative estimation approaches are proposed recently to improve the estimation quality with various assumptions and trade-offs (Chen et al., 2018; Malinin and Gales, 2018; Mandelbaum and Weinshall, 2017). In this work, we use the maximum softmax probability because it is popular, simple to implement, and does not incur much overhead. It is possible to use other uncertainty estimation approaches but it might introduce additional implementation difficulties. For example, the back-propagation for uncertainty loss requires that all forward passes of the Monte Carlo dropout are done. This will drastically increase the memory consumption to save the intermediate results of each pass.

The general idea of uncertainty-aware training is explored for image classification in (Geifman and El-Yaniv, 2019). The SelectiveNet in (Geifman and El-Yaniv, 2019) adds a selection head and an auxiliary head to the network and uses a weighted sum of loss terms to train the networks for a specific coverage. In addition to the different problem settings, our method differs a few ways. Firstly, we avoid training a new head after the deconv layer to do a regression task on all instances. Secondly, SelectiveNet is optimized for a specific coverage and will need to switch to different models for different target coverage. In contrast, we train a unified model that is optimized for overall performance that can be used across all target coverage and the specific coverage in the inference stage can be adjusted smoothly in real-time. (Kumar et al., 2018) proposes a differentiable proxy for the calibration error and optimizes it in the training process. However, the calibration error has fundamental differences with the selective prediction and thus does not imply any benefit to selective segmentation (Ding et al., 2019).

### A.2. Selective segmentation

In selective segmentation for medical images, the referred part can be manually annotated by radiologists or be re-segmented by a DNN based on radiologists' feedback such as scribbles (Wang et al., 2018b). Both of them benefit from a better training objective that suits the nature of selective prediction, which is the focus in this paper. Selective segmentation is closely related to interactive segmentation (Acuna et al., 2018; Xu et al., 2016) but has

distinct objectives. Interactive segmentation is typically done by having a human annotator to check the segmentation results and refining the segmentation based on the human annotator's feedback. The previous research focuses on maximizing the performance of the proposed segmentation and minimizing the required interaction (Benenson et al., 2019; Li et al., 2018). Meanwhile, selective segmentation aims to confidently segment a subset of the input without the need of manually checking which would be time-consuming for medical images, especially for 3D input. For the difficult or ambiguous instances, interactive segmentation tries to segment them correctly based on the iterative human annotations while selective segmentation aims to isolate them from the rest.

Selective segmentation belongs to uncertainty-aware referral for general problems. In the context of DNNs, uncertainty-aware referral is applied to natural image classification (Geifman and El-Yaniv, 2017), disease detection (Leibig et al., 2017) and most recently, medical image segmentation (Eaton-Rosen et al., 2018; Nair et al., 2019; Sander et al., 2019). However, all the existing work directly uses the standard model training algorithm which has not considered the discrepancy between the *training target* and the practical target.

## Appendix B. Proof of Theorem 1

**Proof** From the definition of $\gamma$, we see that, $\gamma$ is maximized for any $c \in (0, 1]$ as long as each wrong prediction has higher uncertainty than any correction predictions i.e. $u_j > u_k$ for $\forall u_j \in U_w, u_k \in U_c$. Now consider $\gamma$ as a scoring rule to measure the quality of uncertainty estimation $q_\theta$ towards the ground truth $q$. If $p_\theta = q$, we have $u_j = 1$ for $u_j \in U_w$ and $u_k = 0$ for $u_k \in U_c$ which leads to maximized $\gamma$. On the other hand, consider a case that $u_j = 0.6$ for $u_j \in U_w$ and $u_k = 0.4$ for $u_k \in U_c$, we have $p_\theta \neq q$ but $\gamma$ is still maximized. As a result, $\gamma$ is a proper scoring rule but not a strictly proper scoring rule for uncertainty estimation. Property $(i)$ is proved.

Here we prove the Property $(ii)$ by verification for clarity. With $\gamma$ and $c$, we first divide all samples into the following four cases (1) $x_i \in X_l, s_i = 0$; (2) $x_i \in X_h, s_i = 0$; (3) $x_i \in X_l, s_i = 1$; (4) $x_i \in X_h, s_i = 1$. For simplicity, we denote the number of instances of these four cases as $N_1, N_2, N_3, N_4$ respectively. Then we can easily define $\psi_c, \psi_1, \gamma$, and $c$ as follows.

$$\psi_c = \frac{N_3}{N_1 + N_3} \tag{4}$$

$$\psi_1 = \frac{N_3 + N_4}{N_1 + N_2 + N_3 + N_4} \tag{5}$$

$$c = \frac{N_1 + N_3}{N_1 + N_2 + N_3 + N_4} \tag{6}$$

$$\gamma = \frac{N_2}{N_2 + N_4} \tag{7}$$

By inserting Equation (5 6 7), we have

$$
\begin{aligned}
\frac{\psi_1 - (1-\gamma)(1-c)}{c} &= \frac{\frac{N_3+N_4}{N_1+N_2+N_3+N_4} - \frac{N_4}{N_2+N_4}\frac{N_2+N_4}{N_1+N_2+N_3+N_4}}{\frac{N_1+N_3}{N_1+N_2+N_3+N_4}} \\
&= \frac{N_3}{N_1 + N_3} \\
&= \psi_c
\end{aligned}
\tag{8}
$$

$\gamma$ has the constraint that the number of correct prediction and the number of wrong predictions in $X_h$ can not exceed the number of correct prediction and the number of wrong predictions in $X$ respectively. Formally:

$$(1-\gamma)(1-c) \leq \psi_1 \tag{9}$$

$$\gamma(1-c) \geq 1-\psi_1 \tag{10}$$

With simple rearrangement we get the constraint $(1-\gamma)(1-c) \in [\psi_1 - c, \psi_1]$. Therefore, property $(ii)$ is proved.

For the property $(iii)$, we have $\frac{\partial \psi_c}{\partial \gamma} = \frac{\gamma(1-c)}{c} > 0$ and $\frac{\partial \psi_c}{\partial \psi_1} = \frac{1}{c} > 0$. ∎

## Appendix C. Implementation Details

The MM-WHS dataset contains 20 CT volumes in the training set. The inputs are 3D patches in the size of 64×64×64. The average results of 5-fold cross-validation is reported. The patch used for GlaS is 192×192. The test dataset is divided into Part A and Part B with the benign and malignant gland. The average results of two parts are reported. We use standard pre-processing and data-augmentation following related work (Sirinukunwattana et al., 2017; Zhuang and Shen, 2016).

The batch size used is 8 and the margin $m$ is set to 0.1. Because $\mathcal{L}_{uncertainty}$ is small in value, we scale it by 1000 to make it comparable with the other term. The optimizer used is Adam with a learning rate set as 0.0002. For the sub-sampling, we sample fixed numbers of instances from $U_w$ and $U_c$ respectively to have a consistent computation overhead in all experiments.

One choice to be made is whether the manual annotation part should be included in the calculation of Dice. We find that the number of segmented instances varies significantly for different classes e.g. 99.8% instances of myocardium were referred when the coverage $c = 0.5$. When the number of instances is too small, the Dice score of the class together with the average Dice fluctuates heavily. Therefore, we choose to include the manual annotation part in the calculation of Dice. Specifically, we assume the referred instances are correctly annotated by radiologists and compute the final Dice score on the $X$ instead of $X_l$. Note that this matches the real use cases and does not change the relative relationship of different methods' Dice.

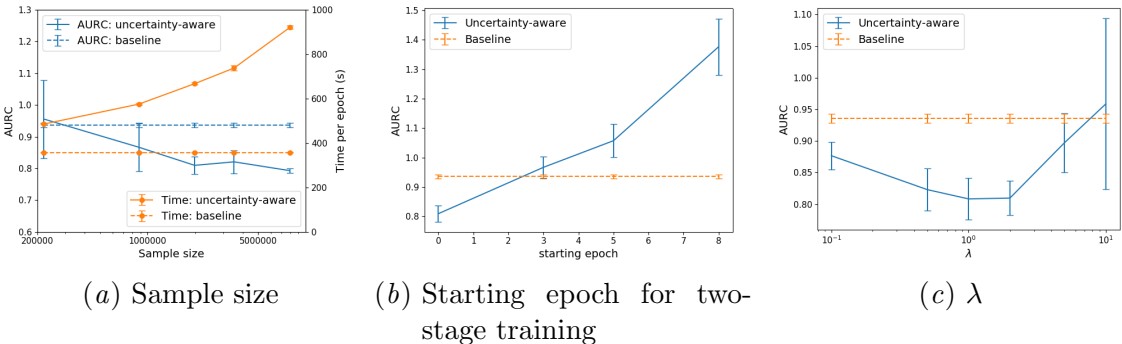

$(a)$ Sample size $\qquad$ $(b)$ Starting epoch for two-stage training $\qquad$ $(c)$ $\lambda$

Figure 5: Model performance comparison with varying hyper-parameters on MM-WHS. Note that AURC is the lower the better.

## Appendix D. Effect of Hyper-parameters

**Effect of sample size.** We first explore the effect of sample size on the training time and the model performance. The training time is measured on a server with an Nvidia 1080 Ti GPU card. The sample sizes tested are $100 \times 2200$, $200 \times 4400$, $300 \times 6600$, $400 \times 2200$, and $600 \times 13200$ in the format of $|U'_w| \times |U'_c|$. The ratio $\frac{1}{22}$ is $\frac{|U_w|}{|U_c|}$ of a normally trained model. As shown in Figure $5(a)$, a small sample size leads to lower performance and higher variance. Since a big sample size does not bring too many benefits, a sample size of $300 \times 6600$ (third point from the left) is used in all following experiments. In this case, our method reduces the AURC by 13.5% at the cost of increased training time.

**Effect of two-stage training.** As shown in Figure $5(b)$, different starting epochs for the second stage is the x-axis. The model is trained for 10 epochs in total. The results show that the performance degrades significantly when we only train the model with $\mathcal{L}_{uncertainty}$ for a smaller number of epochs. The reason is that after the change of loss from $\mathcal{L}_{segmentation}$ to $\mathcal{L}_{u\text{-}seg}$, the model needs enough training to converge. Therefore, we use $\mathcal{L}_{u\text{-}seg}$ from the beginning of the model training in all other experiments.

**Effect of $\lambda$.** As discussed in Section 2, the performance $\psi_c$ is determined by $\psi_1$ and $\gamma$ and $\lambda$ determines the balance between $\psi_1$ and $\gamma$. We explore several different values of $\lambda$ and the results are shown in Figure $5(c)$. The trend of the curve shows that the performance degrades when $\lambda$ is either too big or too small. Additionally, the variance increases with bigger $\lambda$ due to the subsampling. Based on this trend, $\lambda = 2$ is used in all other experiments.

## Appendix E. Per-image Comparison

The per-image comparison between the baseline and our method under different coverage $c$ on GlaS is shown in Figure 6. In order to show our model's per-image effect more thoroughly, we further randomly sample images that has approximately the same per-image coverage (the dots around y axis in Figure 2 and Figure 6) and compare the Dice in Table 2 and Table 3 for MM-WHS and GlaS respectively. As can be seen from the table, our method in general provides better per-image results.

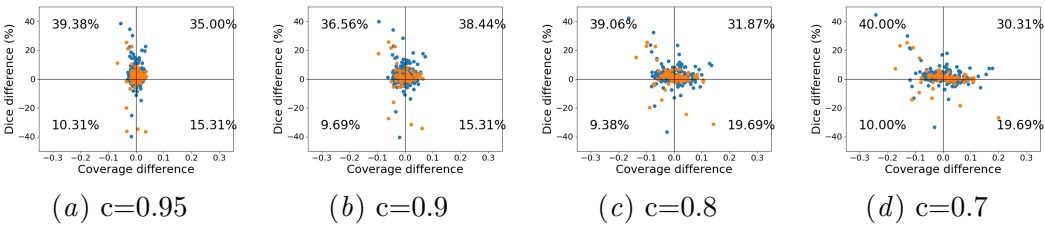

Figure 6: Per-image comparison of Dice and coverage difference under different coverage on GlaS. Blue dots are benign gland images Part A of the test set, orange dots are malignant gland images in Part B of the test set.

## Appendix F. Dice Improvement at Different Coverage

If the improvement on Dice comes from the regularization effect only, then we would expect to see that the Dice difference between our method and the baseline monotonically decreases as coverage reduces. This is because the uncertain parts which are prone to error are continuously removed, and thus the largest possible benefit margin also decreases. When the remaining instances are almost all easy and confidently predicted, different models will have the same performance despite different uncertainty estimation. The Dice difference between 1.00 and 0.95 is shown in Table 4. The Dice difference first increases as the coverage decreases before it starts to decrease. This should come from better uncertainty estimation. MM-WHS is an easier dataset and thus the difference decreases earlier.

Table 2: Per-image comparison of Dice and coverage on MM-WHS.

| Coverage | | Dice (%) | |
|---|---|---|---|
| Baseline | Ours | Baseline | Ours |
| 0.948 | **0.950** | 92.91 | **98.03** |
| 0.949 | **0.950** | **98.14** | 98.06 |
| **0.951** | 0.949 | 94.72 | **98.07** |
| 0.948 | 0.948 | 92.94 | **98.10** |
| **0.901** | 0.900 | **98.24** | 96.71 |
| 0.899 | **0.900** | 98.22 | **98.34** |
| 0.899 | 0.899 | 97.83 | **98.21** |
| **0.902** | 0.899 | 97.92 | **98.37** |
| 0.800 | **0.801** | 99.18 | **99.95** |
| **0.800** | 0.799 | 98.62 | **99.44** |
| 0.799 | **0.802** | 99.60 | **99.95** |
| 0.802 | 0.802 | 98.42 | **99.97** |
| 0.701 | **0.702** | 99.65 | **99.93** |
| 0.701 | 0.701 | **99.96** | 99.27 |
| **0.700** | 0.698 | 99.62 | **99.99** |
| **0.700** | 0.699 | 98.81 | **99.99** |

Table 3: Per-image comparison of Dice and coverage on GlaS.

| Coverage | | Dice (%) | |
|---|---|---|---|
| Baseline | Ours | Baseline | Ours |
| 0.951 | 0.951 | 82.99 | **83.70** |
| **0.950** | 0.949 | 88.10 | **89.30** |
| 0.949 | **0.952** | 87.21 | **87.39** |
| 0.949 | **0.951** | **93.20** | 92.73 |
| 0.897 | **0.898** | 85.96 | **86.79** |
| 0.896 | **0.899** | 89.81 | **91.22** |
| 0.899 | **0.900** | **94.44** | 94.32 |
| 0.897 | **0.900** | 81.51 | **88.09** |
| **0.802** | 0.794 | 87.04 | **93.52** |
| **0.802** | 0.795 | 92.58 | **93.51** |
| 0.795 | **0.798** | 92.17 | **94.60** |
| 0.803 | **0.807** | **94.16** | 93.04 |
| **0.701** | 0.697 | 95.84 | **96.24** |
| 0.702 | **0.708** | 94.37 | **95.07** |
| 0.692 | **0.698** | 92.03 | **93.17** |
| **0.703** | 0.698 | 90.81 | **91.53** |

Table 4: Dice difference at different coverage.

| Coverage | 1.00 | 0.995 | 0.99 | 0.98 | 0.97 | 0.96 | 0.95 |
|---|---|---|---|---|---|---|---|
| MM-WHS | 0.83 | 0.94 | 0.96 | 0.95 | 0.92 | 0.89 | 0.86 |
| GlaS | 1.82 | 1.87 | 1.99 | 2.10 | 2.13 | 2.16 | 2.18 |

