# OpenReview forum: "Uncertainty-Aware Training of Neural Networks for Selective Medical Image Segmentation"
_MIDL.io/2020/Conference — MIDL 2020_

### Official Review · AnonReviewer2 · 2020-03-01
**The authors designed an interesting uncertainty-aware method for semantic segmentation and tested on cardiac and gland images.**

**Rating:** 4
**Confidence:** 4
**Recommendation:** Oral

**Summary:**

1. The authors designed an interesting uncertainty-aware method for semantic segmentation and tested on cardiac and gland images.

2. The paper is well-written with some descriptions in Appendix.

3. Experiments results are significance and comparison results seem good.

4. The proposed method is novel.

**Strengths:**

1. The authors designed an interesting uncertainty-aware method for semantic segmentation and tested on cardiac and gland images.

2. The paper is well-written with some descriptions in Appendix.

3. Experiments results are significance and comparison results seem good.

4. The proposed method is novel.

**Weaknesses:**

1. Some details of the method is missing.

2. The reference included for the MM-WHS segmentation challenge was wrong.

Zhuang, Xiahai, et al. "Evaluation of algorithms for Multi-Modality Whole Heart Segmentation: An open-access grand challenge." Medical image analysis 58 (2019): 101537.

**Detailed Comments:**

The authors designed an interesting uncertainty-aware method for semantic segmentation and tested on cardiac and gland images. The paper is well-written, I just have some suggestions:

1. Is there an automated and adaptive method to determine parameter c? Please elaborate more details.

2. The experiments have been done on MM-WHS challenge and please refer to the correct reference:

Zhuang, Xiahai, et al. "Evaluation of algorithms for Multi-Modality Whole Heart Segmentation: An open-access grand challenge." Medical image analysis 58 (2019): 101537.

3. Are the results in Table 1 got statistical significance?

4. The other concern is that how the proposed framework can cope with the real clinical studies?

**Justification Of Rating:**

The authors designed an interesting uncertainty-aware method for semantic segmentation and tested on cardiac and gland images. The paper is well-written, I just have some suggestions:

1. Is there an automated and adaptive method to determine parameter c? Please elaborate more details.

2. The experiments have been done on MM-WHS challenge and please refer to the correct reference:

Zhuang, Xiahai, et al. "Evaluation of algorithms for Multi-Modality Whole Heart Segmentation: An open-access grand challenge." Medical image analysis 58 (2019): 101537.

3. Are the results in Table 1 got statistical significance?

4. The other concern is that how the proposed framework can cope with the real clinical studies?

**Paper Type:**

methodological development

**Questions To Address In The Rebuttal:**

The authors designed an interesting uncertainty-aware method for semantic segmentation and tested on cardiac and gland images. The paper is well-written, I just have some suggestions:

1. Is there an automated and adaptive method to determine parameter c? Please elaborate more details.

2. The experiments have been done on MM-WHS challenge and please refer to the correct reference:

Zhuang, Xiahai, et al. "Evaluation of algorithms for Multi-Modality Whole Heart Segmentation: An open-access grand challenge." Medical image analysis 58 (2019): 101537.

3. Are the results in Table 1 got statistical significance?

4. The other concern is that how the proposed framework can cope with the real clinical studies?

**Special Issue:**

yes

---

> ### Author Response · Authors · 2020-03-28
> **Response to Reviewer2**
>
> We thank you for your positive and constructive feedback.
>
> (1) The simplest way to determine c is by looking at the risk-coverage curve with the desired risk level with some safety margin. As mentioned in Sec. 2.1, there is also an automated and adaptive method available to get c. The Selection with Guaranteed Risk (SGR) algorithm in (Geifman and El-Yaniv, 2017) finds a proper threshold t (which in turn determines c) to achieve the desired performance level with a certain probability. Although it was proposed in a classification setting, it can be applied to segmentation by considering each image as the input sample and taking segmentation quality as the classification loss.
>
> (2) We are sorry for the reference error. It has been replaced with the correct one.
>
> (3) We did the unpaired two-sample t-test and found that there is a statistical significance (two-tailed, p=0.0002/0.0401 for MM-WHS/GlaS).
>
> (4) In clinical practice, the uncertainty-augmented segmentation can be used to improve human-machine collaboration (Nair et al., 2019). The entire segmentation results by the model are presented to radiologists with the uncertainty information (e.g., through color coding for different levels of uncertainty). Instead of checking the entire image carefully including non-interested contours/regions, radiologists can only pay special attention to the uncertain part. This can be useful for example when the segmentation results need to be read by radiologists for further diagnosis, or in the case of manual annotation assistance by radiologists is needed.

---

### Official Review · AnonReviewer3 · 2020-03-10
**Interesting idea but difficult to read**

**Rating:** 2
**Confidence:** 3

**Summary:**

The paper presents a novel method for selective segmentation that tries to maximize the performance on a practical target instead of the full training target by introducing an uncertainty loss. The proposed training scheme can be applied to most existing segmentation frameworks in a plug-and-play manner. The method was evaluated two datasets (MM-WHS and GlaS) and outperformed the baseline (without the proposed uncertainty loss) in all metrics.

**Strengths:**

1) By focusing on uncertainty, this work addresses an important direction in medical image segmentation. Unlike many other works related to uncertainty in medical image segmentation, the paper introduces a new principle, selective segmentation, which is borrowed from the classification literature.

2) The method is evaluated extensively. The paper demonstrates the benefit of the method on different metrics, two datasets, and several experiments. The multitude of experiments helps the reader to get a better understanding of the approach. Also, the hyperparameter analysis is quite helpful.

3) The problem and main terms are well-introduced (especially Figure 1) and are beneficial for the general understanding.

4) The authors provide code. I consider this is very important when introducing a new method because it improves reproducibility. Unfortunately, there are still a lot of papers presenting new methods without code.

**Weaknesses:**

1) In my opinion, the main weakness of the paper is the writing and the lack of clear messages. In detail, this leads to the following problems:
a) Difficult to read. The paper lacks a description of the idea in simple words easy to understand. Although the method seems valid, I find it hard to follow all the details in section 2.2. An improved structure, including repetitions of the important information, would be beneficial. An example is the description of $\gamma$ (including the theorem), which is described in detail. However, the final loss does not contain $\gamma$ because of its non-differentiability. I believe this could be simplified.
b) The motivation for the uncertainty loss is unclear. The softmax cross-entropy is described as a proper scoring rule which tries to recover the actual distribution $q$. It is also stated that selective segmentation does not require recovering the distribution $q$. It is unclear why one should not recover the actual $q$ (even though not required) with the cross-entropy loss if this loss is anyway used to optimize the segmentation task. Or, why is the uncertainty loss even needed if the cross-entropy is already trying to do more than required?
c) The benefit is not obvious. Although the experiments help to understand the method better, the benefits of the proposed method are not obvious. It seems that the initial Dice coefficient performance already improves (although c=1 not shown in Table 1) with the proposed method. However, it is not clear whether the improved performances at the subsequent coverage values (e.g., 0.95, 0.9, …) are due to an improved uncertainty or the initial benefit. It seems that the baseline has higher deltas between two consecutive coverage values. Additional clarifications of the results and a more extensive discussion of the results are required to improve the understanding.

2) The adoption of the proposed setup is limited. As described in the introduction, selective segmentation only predicts voxels that the model is certainty about, and the remaining are left for expert annotation (Figure 1). This setup is, in my opinion, not realistic. If a radiologist has to annotate all uncertain voxels in an image, the time gain compared to full manual annotation will most likely be very limited.

**Detailed Comments:**

Minor:
- It is unclear how the uncertainty loss is obtained from $\gamma$. It could be helpful to clarify why the non-differentiability of $\gamma$ results in the uncertainty loss.
- In the supplementary material is written that $\lambda=2$ has been used, but at the same time, it is also mentioned (in Appendix C.) that the uncertainty loss is increased by a factor of 1000. Isn’t $\lambda=2000$, then?
- For a continuation of this work, it might also be interesting to adapt/weight the segmentation loss with the level of uncertainty.

Typos:
- “actical target” instead of “practical target” (2 times)
- In the introduction, the authors write: “More importantly, we show that the practical target can be decomposed into the training target and a novel uncertainty target”. Shouldn’t it rather be: "... we show that the training target can be decomposed into the practical target ..." training target can be decomposed into the practical target and a novel uncertainty target” (according to Figure 1)?

**Justification Of Rating:**

The paper provides an interesting approach and extensive evaluation. Unfortunately, the structure and writing make the paper hard to read and understand. Therefore, I suggest rejection of the paper unless the readability is improved.

**Paper Type:**

methodological development

**Questions To Address In The Rebuttal:**

I would like the authors to address the points 1a-1c of the weaknesses above.

**Special Issue:**

no

---

> ### Author Response · Authors · 2020-03-28
> **Response to Reviewer3**
>
> We thank you for your time and thoughtful review.
>
> (a) We have carefully revised the manuscript following your suggestions to improve the clarity. Specifically, we have rewritten some descriptions and divide Section 2.2 into two sections to clearly separate the description around \gamma and the practical uncertainty loss. Regarding the message of Sec 2.2, the description of \gamma is an analysis of the problem in a rigorous manner, which is a key contribution of this paper. We have shown that \gamma is the right objective to transfer the original loss function to a suitable one for selective segmentation. However, because it is not clear how to directly optimize \gamma, we resort to an approximation to it. We discussed how \gamma is related to the uncertainty loss used after Equ. 2 (page 5). Given the description of \gamma, it is possible for follow up research to find a better approximation to \gamma, which is left for future work.
>
> (b) It is true that if the cross-entropy loss can recover the ground truth distribution q then no uncertainty loss is needed. However, giving the limited capability of the model and practical issues, using cross-entropy loss does not ensure perfect recovery of q. As a result, an uncertainty loss is needed to work as complementary to the cross-entropy. When used together, the final loss is more aligned with our practical target which improves the performance. As an analogy, when a student gets a full score by solving 5 problems out of a total of 10 in a test. Solving all 10 problems would not hurt, but when the time is not sufficient, it is better to focus on the easiest 5 of them instead of all problems. The same idea has motivated related research in the classification scenario (Geifman and El-Yaniv, 2019).
>
> (c) As both our method and the baseline use the same network (with different loss functions), the initial benefit at c=1.0 is a regularization effect of the new loss function which does not penalize heavily for the low confidence wrong prediction. If the benefit comes from the regularization effect only, then we would expect to see that the Dice difference between our method and the baseline monotonically decreases as coverage reduces. This is because the uncertainty parts which are prone to error are continuously removed, and thus the largest possible benefit margin also decreases. When the remaining instances are almost all easy and confidently predicted, different models will have the same performance despite different uncertainty estimation. However, when we look at the Dice difference between 1.00 and 0.95 as shown below.
>
> Coverage:   1.00  0.995  0.99  0.98  0.97  0.96  0.95
> MM-WHS:   0.83  0.94    0.96  0.95  0.92  0.89  0.86
> GlaS:            1.82  1.87    1.99  2.10  2.13  2.16  2.18
>
> We can see that the Dice difference first increases as the coverage decreases before it starts to decrease. This should come from better uncertainty estimation. MM-WHS is an easier dataset and thus the difference decreases earlier.

---

> > ### Comment · AnonReviewer3 · 2020-04-03
> > **Good clarifications**
> >
> > Thank you for your response. My main concern was the clarity of the method section. The authors seem to have improved this part by dividing section 2.2 into two sections and by rewriting some descriptions. Although I cannot verify the improvement, I assume the authors addressed my main concern and will consequently update my score.
> >
> > I find the results in (c) quite helpful. Please consider adding them as supplementary material.

---

### Official Review · AnonReviewer1 · 2020-03-15
**Solid paper on "plug-and-play" way of imposing uncertainty as a part of learning scheme.**

**Rating:** 4
**Confidence:** 4
**Recommendation:** Oral, Poster

**Summary:**

The paper presents a model-independent approach to consider uncertainty which consequently helps the overall segmentation performance. It provides several concepts of uncertainty to consider during training and shows a proxy loss function to explicitly account for it. It overall provides a nice set of experiments and analyses for interesting takeaway messages regarding uncertainty.

**Strengths:**

1. Well written and easy to read.
2. The distinction between the types of scoring rules is appreciated.
3. Extensive experiments with thorough analyses.
4. Consistent improvement across several experimental setups.

**Weaknesses:**

I do not have major comments on weaknesses. A minor comment is on self-containedness with the uncertainty map figure not being in the main paper. This is quite minor though. Another minor comment is the lack of mentioning of existing uncertainty estimation methods (e.g., MC-dropout) which could output a different type of uncertainty which may replace the softmax.

**Justification Of Rating:**

It is an overall solid paper with well-written details and thorough experiments. The method is simple and reasonable, although I wonder if the softmax is the only uncertainty measure it could consider. Still, there are interesting observations from the experimental analyses that may benefit readers.

**Paper Type:**

both

**Questions To Address In The Rebuttal:**

1. It was not clear in 2.1. whether the threshold t splits images (i.e., 3D volumes) or voxels, since the threshold seemed to be applied for the individual voxel-level uncertainty u_i. I think each "instance" is still a voxel. If so, how could radiologists do manual segmentation? Each image could have quite isolated voxels in X_l and X_h.


**Special Issue:**

yes

---

> ### Author Response · Authors · 2020-03-28
> **Response to Reviewer1**
>
> Thanks for your careful review and thoughtful comments!
>
> (1) We actually mentioned the existing uncertainty estimation methods in Appendix A.1, but we agree that the possibility of using alternative uncertainty estimation methods, which is a promising future research topic, worth more discussion. We note that the back-propagation for uncertainty loss requires all forward passes of the MC-dropout are done and this will drastically increase the memory consumption to save the intermediate results of each pass. Explicit discussions about this have been added to Appendix A.1. We also moved the uncertainty map figure to the main paper.
>
> (2) For the image split, currently the threshold is applied for individual voxels/pixels following earlier work in this direction (Nair et al., 2019; Sander et al., 2019). In experiments, most of the uncertain/certain pixels are quite clustered. In clinical practice, the proposed framework can be used to improve human-machine collaboration. The entire segmentation results by the model are presented to radiologists with the uncertainty information (e.g., through color coding for different levels of uncertainty). Instead of checking the entire image carefully including non-interested contours/regions, radiologists can only pay special attention to the uncertain part. This can be useful for example when the segmentation results need to be read by radiologists for further diagnosis, or in the case of manual annotation assistance by radiologists is needed.

---

### Meta-Review · Area_Chair1 · 2020-04-06
**MetaReview of Paper102 by AreaChair1**

**Rating:** 4
**Recommendation For Accepted Papers:** Oral

**Metareview:**

The reviewers agree that the proposed method is novel and interesting, and that the results are backed by good experimental results. The questions raised during teh review have been answered well. I thus recomend this paper be accepted.

**Paper Type:**

methodological development

**Special Issue:**

no

---

### Decision · Program_Chairs · 2020-04-11

Accept